# CLMSM: A Multi-Task Learning Framework for Pre-training on Procedural Text

**Abhilash Nandy**[*]   **Manav Nitin Kapadnis**[*]   **Pawan Goyal**   **Niloy Ganguly**
nandyabhilash@kgpian.iitkgp.ac.in       iammanavk@gmail.com
Indian Institute of Technology Kharagpur
India

## Abstract

In this paper, we propose CLMSM, a domain-specific, continual pre-training framework, that learns from a large set of procedural recipes. CLMSM uses a Multi-Task Learning Framework to optimize two objectives - a) Contrastive Learning using hard triplets to learn fine-grained differences across entities in the procedures, and b) a novel Mask-Step Modelling objective to learn step-wise context of a procedure. We test the performance of CLMSM on the downstream tasks of tracking entities and aligning actions between two procedures on three datasets, one of which is an open-domain dataset not conforming with the pre-training dataset. We show that CLMSM not only outperforms baselines on recipes (in-domain) but is also able to generalize to open-domain procedural NLP tasks.

## 1   Introduction

Procedural text consists of a set of instructions that provide information in a step-by-step format to accomplish a task. Procedural text can be found in different domains such as recipes (Marin et al., 2018; Bień et al., 2020; Chandu et al., 2019; Majumder et al., 2019; Bosselut et al., 2017), E-Manuals (Nandy et al., 2021), scientific processes (Mishra et al., 2018), etc. Procedural reasoning requires understanding the interaction between participating entities and actions associated with such entities in procedures. Recent works have made attempts at tracking entities (Mishra et al., 2018; Dalvi et al., 2019; Faghihi and Kordjamshidi, 2021) and aligning actions (Donatelli et al., 2021) across the different steps of procedures to evaluate procedural reasoning. The performance of these tasks can possibly be enhanced by specialized pre-training on large procedural corpora, such that the pre-training objectives are aligned with the downstream tasks at hand. However, none of the prior art has studied

the effects of pre-training to solve such procedural NLP tasks thoroughly.

Current self-supervised pre-training techniques such as Masked Language Modeling (MLM) (Liu et al., 2019; Devlin et al., 2018), Next Sentence Prediction (NSP) (Devlin et al., 2018), text denoising (Raffel et al., 2020; Lewis et al., 2019), etc. have been shown to perform well on downstream NLP tasks such as Question Answering (QA), Named Entity Recognition (NER), Natural Language Inference (NLI), etc. While such methods could capture local contexts proficiently (Lai et al., 2020; Sun et al., 2021), these methods fail in capturing procedural, step-wise contexts required for tracking entities and aligning actions in procedural texts (Mishra et al., 2018; Dalvi et al., 2019) as the learning objectives do not consider the context of the entire procedure, and the interaction between entities and actions.

To address these challenges, in this paper, we propose CLMSM[1], which is a combination of *Contrastive Learning* (CL) using domain-specific metadata, and a self-supervised *Masked Step Modeling* (MSM) to perform domain-specific continual pre-training for procedural text. We use CL with hard triplets (Cohan et al., 2020; Ostendorff et al., 2022) which enables learning fine-grained contexts associated with participating entities and actions, by pulling representations of highly similar procedures closer and pushing representations of slightly dissimilar procedures further away from each other. Prior works on instructional video understanding such as Narasimhan et al. (2023) have also shown to benefit from the use of MSM in learning step-wise representations to capture global context of the entire video. Inspired from such works, we incorporate MSM for procedural text. MSM considers a step as a single entity and learns step-wise context, as it predicts tokens of randomly masked step(s), given the other steps of the procedure. We pre-train

---

[*]Equal contribution.

[1]Contrastive Learning cum Masked Step Modeling

our model in the recipe domain, as recipes provide step-wise instructions on how to make a dish, making them an ideal candidate for pre-training.

We experiment with various downstream procedural reasoning tasks that require understanding of the changes in states of entities across steps of the procedure, triggered due to actions carried out in each step. The first task of tracking the movement of constituent entities through a procedure is carried out on the NPN-Cooking Dataset (Bosselut et al., 2017) of the recipe domain, and ProPara Dataset (Mishra et al., 2018) of the open-domain (to evaluate the generalizability of domain-specific pre-training), while the second task is to align actions across different recipes of the ARA dataset (Donatelli et al., 2021). We show that CLMSM performs better than several baselines when evaluated on domain-specific as well as open-domain downstream tasks. Specifically, CLMSM provides performance improvements of 1.28%, 2.56%, and 4.28% on the NPN-Cooking, ProPara, and ARA action alignment datasets, respectively. Our extensive ablation study suggests that a) the proposed multi-task learning framework outperforms using any one of the two objectives or using these objectives sequentially, b) masking an entire step is much more effective than masking a random span, and c) the use of hard triplets during CL is much more effective than easy triplets.

The remainder of the paper is organized as follows: We begin by introducing the proposed pre-training framework in Section 2, followed by the experimental setup in Section 3, and a description of the datasets, fine-tuning, and evaluation frameworks used for the two downstream tasks, along with experiments, results and analysis in Sections 4 and 5. Finally, we conclude with a summary of our findings in Section 6. The code and models used in this paper are available on Github [2].

## 2 Pre-training

CLMSM follows a Multi-Task Learning (MTL) framework, where CL and MSM are performed simultaneously on 3 input procedures per training sample (as shown in Figure 1). In this section, we first describe the CL and MSM methods, followed by a detailed description of the CLMSM framework, ending with the details of pre-training in the domain of recipes.

---

[2]https://github.com/manavkapadnis/CLMSM_EMNLP_2023

### 2.1 Contrastive Learning using procedure similarity (CL)

As shown in Figure 1, we use a Triplet Network (Cohan et al., 2020; Ostendorff et al., 2022), where three procedures serve as inputs for three bidirectional encoders (such as BERT (Devlin et al., 2018), RoBERTa (Liu et al., 2019)), the first (Anchor) and the second (Positive) procedures being similar, and the first (Anchor) and the third (Negative) procedures being dissimilar (based on metadata). Note that we use hard triplets to learn fine-grained similarities and differences in entities and actions across procedures. The encoders have hard parameter sharing (Caruana, 1993). The three encoded representations are used to formulate a triplet margin loss function, denoted by $\mathcal{L}_t$. Mathematically,

$$\mathcal{L}_t = max\{d(P_1, P_2) - d(P_1, P_3) + 1, 0\} \quad (1)$$

where $P_1, P_2, P_3$ refer to the representations of the three procedures, and $d(.,.)$ represents the L2 norm distance.

### 2.2 Mask-Step Modeling (MSM)

Here, we formulate a self-supervised multiple-masked token prediction task. Given a procedure (a procedure comprises multiple steps, a step comprises one or more sentences), tokens of randomly selected $M(\geq 1)$ step(s) are masked and the task is to predict all such masked tokens.

In CLMSM, each input procedure (with tokens of random step(s) masked) is passed through a Bidirectional Transformer. The MSM Loss function $\mathcal{L}_{MSM}$ is the sum of the MLM Loss (Devlin et al., 2018; Liu et al., 2019) over all the masked tokens, for all the $N$ input procedures per training sample (here, $N = 3$, as in Figure 1). Mathematically,

$$\mathcal{L}_{MSM} = \sum_{i=1}^{N} \sum_{j \in \mathcal{S}_i} \mathcal{L}_{MLM}(i, j), \quad (2)$$

where $\mathcal{S}_i$ is the set of masked token indices in the $i^{th}$ input procedure, and $\mathcal{L}_{MLM}(i, j)$ is the MLM Loss for predicting the $j^{th}$ masked token, given the $i^{th}$ input procedure.

### 2.3 CLMSM Framework

CLMSM framework is shown in Figure 1. Here, MSM and CL are both carried out simultaneously in an MTL setting for $K$ epochs, and the Transformer obtained after pre-training is used for fine-tuning. Intuitively, CLMSM learns the step-wise

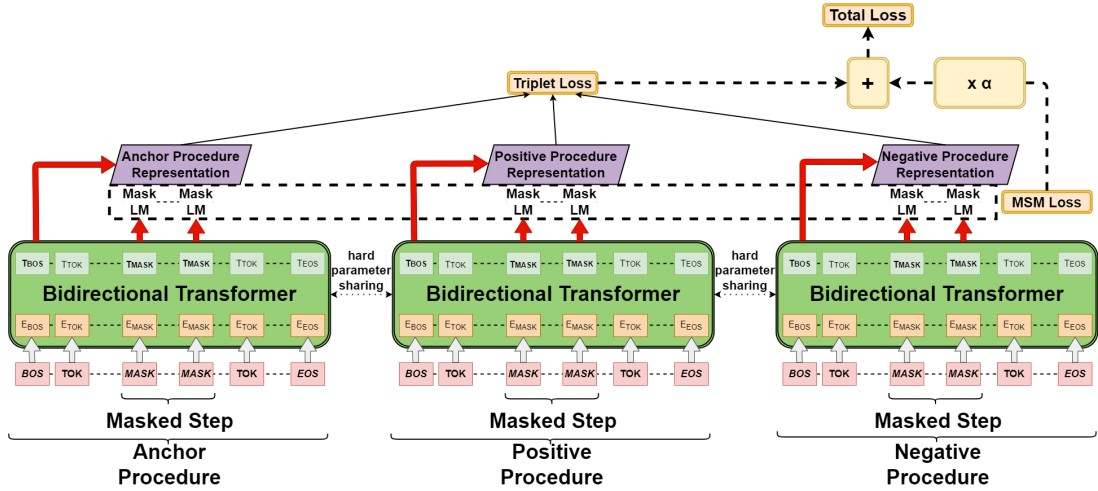

Figure 1: CLMSM pre-training framework: Step-Masking and Contrastive Learning objectives are used. MSM Loss obtained by masking step(s) in each recipe is linearly combined with Triplet Loss, and total loss is backpropagated.

and fine-grained procedural contexts simultaneously via MSM and CL, respectively.

The loss $\mathcal{L}$ (referred to as 'Total Loss' in Figure 1) backpropagated during pre-training is the sum of the triplet margin loss and scaled step masking loss. Mathematically,

$$\mathcal{L} = \mathcal{L}_t + \alpha * \mathcal{L}_{MSM}, \quad (3)$$

where $\alpha$ is a positive scaling constant for $\mathcal{L}_{MSM}$.

## 2.4 Pre-training in the Recipe Domain

The pre-training data contain over 2.8 million recipes from 4 different sources, described in detail *in Section B.4 of Appendix*. We further filter the dataset to remove recipes that contain less than 4 steps, which results in 2, 537, 065 recipes. Since a recipe contains a list of ingredients in addition to the steps (procedure) for making the dish, we modify the recipe by adding the list of ingredients as a step before the original steps of the recipe. We randomly choose from 20 manually curated templates[3] to convert the list of ingredients to a step (E.g. "Collect these items: {ingredients separated by commas}", "Buy these ingredients from the supermarket: {ingredients separated by commas}", etc.). An example of such modification of a recipe is discussed *in Section B.4 of Appendix.*

To apply MSM, the number of steps masked per recipe is taken randomly as $M = 1$ or 2 in our experiments. We propose the following heuristic as a sampling procedure of hard triplets for CL - Given a recipe as an anchor, a triplet is sampled by

considering a different recipe of the same dish as a positive, and a recipe of a "similar" dish as a negative. For a given recipe, a set of recipes of "similar" dishes can be filtered in the following way - (1) We take the given recipe name[4], and find the top 20 most similar recipe names in the corpus based on cosine similarity between pre-trained Sentence Transformer (Reimers and Gurevych, 2019) embeddings. (2) This is followed by filtering the recipe names that have common noun phrase root (for example, if banana **smoothie** is given, strawberry **smoothie** is similar)[5]. We provide an example of anchor, positive, and negative recipes in Figure 2. We observe that (1) most of the ingredients are common, making the anchor and positive highly similar, and the anchor and negative marginally dissimilar, and (2) non-common ingredients with the negative sample are thus discriminative (e.g. Chicken and Mixed vegetables); CL loss helps CLMSM to quickly learn those fine-grained features.

**Pre-training Setup:** Prior to applying CLMSM, the bi-directional transformer is pre-trained on the corpus of recipes via MLM, as in Liu et al. (2019). In CLMSM, we use a batch size of 32 along with AdamW optimizer (Loshchilov and Hutter, 2017) with an initial learning rate of $5 \times 10^{-5}$, which linearly decays to 0. CLMSM is pre-trained using MSM and CL simultaneously on 2 Tesla V100 GPUs for $K = 1$ epoch[6]. We scale down the MSM loss to bring both the MSM and CL losses in the

---

[3]a large number of templates is used to provide variety to this added step across recipes

[4]the term "recipe name" refers to the name of the dish
[5]used chunk.root.text (https://spacy.io/usage/linguistic-features#noun-chunks) - MIT License
[6]Details on number of parameters and compute are discussed *in Section B.4 of Appendix*

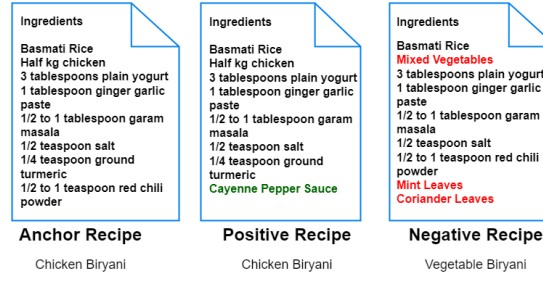

Figure 2: Ingredients of the anchor, positive, and negative recipes of a triplet. Ingredients in black are common in all recipes, those in green and red are non-common ingredients in positive and negative recipes respectively

same range, by setting the coefficient $\alpha$ in Equation 3 (for getting the total loss) to $0.05$.

## 3 Experimental Setup

We evaluate CLMSM on the tasks of **entity tracking** and **recipe alignment**[7]. We assess the performance of CLMSM vis-a-vis baselines and various variations of CLMSM, and list inferences that can be drawn. Note that, from here onwards, the subscript in any model notation represents the encoder used (RB - RoBERTa-BASE, RL - RoBERTa-LARGE, BERT - BERT-BASE-UNCASED).

### 3.1 CLMSM Variations

For all the tasks we consider the following variations of CLMSM for comparison as baselines - (1) **CLMSM(CL)** uses only the CL pre-training objective (2) **CLMSM(MSM)** uses only the MSM pre-training objective (3) **CLMSM(CAS.)** uses a cascaded framework (CAS.) instead of MTL, i.e, it performs CL followed by MSM, instead of optimizing both the objectives simultaneously. (4) **CLMSM(RS)** uses a variation of MSM by masking random spans instead of step(s). This variation is used to establish the utility of step-masking as a supervision signal for procedural text. For a fair comparison, we mask the same percentage of tokens when masking random spans and when masking step(s). (5) **CLMSM(EASY)** uses easy triplets instead of hard triplets to see if fine-grained differences between anchor and negative procedures are indeed helpful. For sampling easy triplets, the positive recipe name is similar to the anchor recipe name (definition of "similar" is mentioned in Section 2.4), and the negative recipe name is not similar to the anchor recipe name.

---
[7]Datasets are in the English Language

## 4 Experimental Details: Tracking Entities

For the task of entity tracking, we evaluate on (1) NPN-Cooking Dataset in the recipe domain to see how helpful domain-specific training is for **in-domain** tasks, and (2) Propara Dataset in the **open-domain** to see how generalizable a domain-specific pre-trained model is.

### 4.1 Problem Definition: Tracking Entity

On execution of a step in a procedure, an **entity** can have three predefined **locations**: non-existence, unknown-location, or known-location, where the "known location" is expressed by a span of text describing where the entity is. At the beginning of the procedure, each entity is assigned a location. Besides the location, each entity is also assigned a **status**. At the step where the entity first appears, it gets a status of **created**. If the location changes, the status changes to **moved** and once the entity ceases to exist, then the status assigned is **destroyed**. An entity that has not been created till now or has been destroyed has no status.

We elaborate on it using an example. The procedural text of photosynthesis is described in Table 1 which contains a list of steps. Each row in the table represents a step in the procedure, with the first column containing the sentences that describe each step, and the remaining columns containing information about entities involved in the process and their locations after each step. The location of the entities "Before the process starts" is their initial location and is not affected by the process. Considering the entity "water", it is in the soil before the procedure begins, and **moves** to the root after execution of the first step, stays in the leaf after the second and third steps. The fourth step converts the water, light, and CO2 into a mixture, and thus, water ceases to exist thereafter. Thus, "water" is **destroyed** and "mixture" is **created** in the fourth step.

Formally, given a procedure $P$ consisting of a list of $n$ steps $P = \{p_1, ..., p_n\}$, the entity tracking task identifies the location $L$ and the status $S$ of entity $e$ at step $p_i \ \forall(i)$. Mathematically, $(S, L) = F(e, p_i)$.

### 4.2 Fine-tuning

The transformer model provides a step-dependent entity representation which is used to output the status and location of the entity. The status is derived using a linear classification module. To predict the location of an entity the approach of Span-Based

| | Entities | | | | |
|---|---|---|---|---|---|
| Procedure | Water | Light | CO2 | Mixture | Sugar |
| (Before the process starts) | Soil (K) | Sun (K) | $\emptyset$ (U) | $\emptyset$ (N) | $\emptyset$ (N) |
| Roots absorb water from soil | Root (K) | Sun (K) | $\emptyset$ (U) | $\emptyset$ (N) | $\emptyset$ (N) |
| The water flows to the leaf | Leaf (K) | Sun (K) | $\emptyset$ (U) | $\emptyset$ (N) | $\emptyset$ (N) |
| Light from the sun and CO2 enter the leaf | Leaf (K) | Leaf (K) | Leaf (K) | $\emptyset$ (N) | $\emptyset$ (N) |
| Water, light, and CO2 combine into a mixture | $\emptyset$ (N) | $\emptyset$ (N) | $\emptyset$ (N) | Leaf (K) | $\emptyset$ (N) |
| Mixture forms sugar | $\emptyset$ (N) | $\emptyset$ (N) | $\emptyset$ (N) | $\emptyset$ (N) | Leaf (K) |

Table 1: An annotated sample from the ProPara dataset. "$\emptyset$" means the location of the entity is undefined. The location of the entity in each step is represented within brackets ('K' - known-location, 'U' - unknown-location, 'N' - non-existence)

Question Answering is followed, whereby a location span is output. Sum of Cross-entropy losses pertaining to the location and status prediction is used to fine-tune the model.

The fine-tuning is carried out in two stages as recommended by Faghihi and Kordjamshidi (2021) - first on the SQuAD 2.0 Dataset (described *in Section D.2 of Appendix*), and then on one of the two target datasets. The hyper-parameters used for fine-tuning are described *in Section D.2 of Appendix*.

### 4.3 Common Baselines

The following baselines have been shown to work satisfactorily well for both in-domain and open-domain downstream tasks in prior works, and are hence common for both ProPara and NPN-Cooking Datasets - **KG-MRC** (Das et al., 2018), **DYNAPRO** (Amini et al., 2020), and **RECIPES$_{RB}$** (model obtained after pre-training RoBERTa-BASE on the corpus of recipes using MLM, fine-tuned as described in Section 4.2.). The baselines are described in detail *in Section D.3 of Appendix.*

### 4.4 Entity Tracking on NPN-Cooking Dataset

The NPN-Cooking[8] dataset (Bosselut et al., 2017) consists of $65,816$ training, $175$ development, and $700$ evaluation recipes[9]. The dataset consists of Human Annotations corresponding to changes in the status and location of the entities throughout the procedure.

### 4.4.1 Evaluation Metrics

The model is evaluated on the accuracy of the predicted locations at steps where the location of the ingredients change. The status prediction compo-

---

[8]**NPN** stands for **N**eural **P**rocess **N**etworks

[9]This dataset is a subset of the larger "Now You're Cooking" dataset (Kiddon et al., 2016).

nent of the model is not employed in this evaluation (as in Faghihi and Kordjamshidi (2021)).

### 4.4.2 Dataset-specific Baseline

We use Neural Process Network Model (**NPN-Model**) (Bosselut et al., 2017) as a baseline, described in detail *in Section D.4.1 of Appendix.*

### 4.4.3 Results

We compare CLMSM$_{RB}$ (pre-trained CLMSM model with RoBERTa-BASE ($RB$) encoder) with the Baselines and CLMSM Variations. The results are shown in Table 2.

**Comparison with Baselines:** We can infer that - (1) CLMSM$_{RB}$ gives significantly better test accuracy ($p < 0.05$) than all baselines, performing $1.28\%$ better than the best baseline ($RECIPES_{RB}$). This suggests that the pre-training objectives are helpful. (2) $RECIPES_{RB}$ is the best baseline due to domain-specific pre-training (3) DYNAPRO uses an end-to-end neural architecture and a pre-trained language model to get entity representations and performs better than NPN-Model and KG-MRC, but performs worse than $RECIPES_{RB}$ due to lack of in-domain knowledge before fine-tuning. (4) NPN-Model (which is a data-specific baseline) and KG-MRC give poor results due to the lack of an end-to-end neural architecture and prior knowledge through pre-trained models.

**Comparison with CLMSM Variations:** We can infer that - (1) CLMSM$_{RB}$ performs better than CLMSM$_{RB}(CAS.)$, CLMSM$_{RB}(RS)$, and CLMSM$_{RB}(EASY)$, suggesting that performing MTL on CL (using hard triplets) and MSM objectives is indeed helpful for downstream in-domain Enity Tracking (2) CLMSM$_{RB}$ performs only slightly inferior compared to when using the objectives individually in CLMSM$_{RB}(CL)$ and CLMSM$_{RB}(MSM)$, with a minimal drop of $0.78\%$ and $1.12\%$ respectively. As pre-training and fine-tuning are performed in the same domain, the simple framework of using a single loss is sufficient. We shall see later that in case of ProPara, where pre-training and fine-tuning are carried out in different domains, combination of the two losses in CLMSM is helpful in enhancing performance.

### 4.5 Entity Tracking on ProPara Dataset

The ProPara Dataset (Mishra et al., 2018) consists of $488$ human-authored procedures (split 80/10/10 into train/dev/test) with $81k$ annotations regarding

|  | TEST ACCURACY |
|---|---|
| NPN-Model | 51.3 |
| KG-MRC | 51.6 |
| DYNAPRO | 62.9 |
| RECIPES$_{RB}$ | 63.8 |
| CLMSM$_{RB}(CL)$ | 65.13 |
| CLMSM$_{RB}(MSM)$ | **65.35** |
| CLMSM$_{RB}(CAS.)$ | 63.99 |
| CLMSM$_{RB}(RS)$ | 63.46 |
| CLMSM$_{RB}(EASY)$ | 63.68 |
| CLMSM$_{RB}$ | 64.62 |

Table 2: Results when fine-tuned on the NPN-Cooking Dataset - Baselines and CLMSM Variations vs. CLMSM

the changing states (existence and location) of entities in those paragraphs, with the goal of predicting such changes. We focus on evaluating the **entity state tracking** task in this work.

### 4.5.1 Evaluation Metrics

The model is evaluated three-ways corresponding to a given entity $e$. They are: **Category 1**: which of the three transitions - created, destroyed, or moved undergone by an entity $e$ over the lifetime of a procedure; **Category 2**: the steps at which $e$ is created, destroyed, and/or moved; and **Category 3**: the location (span of text) which indicates $e$'s creation, destruction or movement. We also report the average of the 3 evaluation metric scores.

### 4.5.2 Dataset-specific Baselines

We use the following dataset-specific baselines - a **rule-based** method called **ProComp** (Clark et al., 2018), a **feature-based** method (Mishra et al., 2018) using Logistic Regression and a CRF model, **ProLocal** (Mishra et al., 2018), **ProGlobal** (Mishra et al., 2018), **EntNet** (Henaff et al., 2016), **QRN** (Seo et al., 2016), **NCET** (Gupta and Durrett, 2019), and **RECIPES$_{RL}$** (which is similar to **RECIPES$_{RB}$**, except that, RoBERTa-LARGE is used instead of RoBERTa-BASE.). The baselines are described in detail ***in Section D.5.1 of Appendix.***

### 4.5.3 Results

We compare CLMSM$_{RB}$ and CLMSM$_{RL}$ (CLMSM with an encoder having RoBERTa-LARGE architecture)[10] with Baselines and CLMSM Variations. The results are shown in Table 3. Additionally, we compare CLMSM with much larger LLMs (Large Language Models).

---

[10]We use RoBERTa-LARGE based models here similar to Faghihi and Kordjamshidi (2021)

**Comparison with Baselines:** We can infer that - (1) CLMSM$_{RL}$ performs the best w.r.t Cat. 1, Cat. 2, and average Scores, even though CLMSM is pre-trained on domain-specific corpora (which is different from the domain of the ProPara Dataset). CLMSM$_{RL}$ gives a significant improvement ($p < 0.05$) of $2.56\%$ over the best baseline (DYNAPRO) in terms of average category score. (2) CLMSM$_{RB}$ outperforms 5 baselines on all 3 metrics, and gives better Average Score than 8 baselines. (3) NCET and DYNAPRO achieve high evaluation metrics, as NCET uses extra action/verb information, while DYNAPRO uses an open-domain pre-trained language model in its framework. This is the reason behind these models performing better than $RECIPES_{RB}$, which is pre-trained on recipes. DYNAPRO especially gives a high Cat. 3 Score due to an additional Bi-LSTM that smoothens predicted locations across steps, while adding additional fine-tuning overhead. (4) KG-MRC performs well, as it builds a dynamic Knowledge Graph of entities and states, which then becomes a structured input to the MRC, making it easier to figure out the entity's state. (5) EntNet and QRN perform poorly, as they both follow very simple recurrent update mechanisms, that are not complex enough to understand procedures effectively. (6) ProGlobal shows a considerable improvement in the Cat. 3 Score, as it attends across the entire procedure, thus having better global consistency. (7) ProLocal performs slightly better, but the improvement in the Cat. 3 Score (corresponding to the location of an entity) is negligible. This can be due to ProLocal making local predictions, thus having poor global consistency. (8) Rule-based and Feature-based baselines perform poorly, as they cannot generalize well to the unseen vocabulary in the test set.

**Comparison with CLMSM Variations:** (1) CLMSM$_{RB}$ gives the best Average Score, giving an improvement of $1.73\%$ compared to the best variant CLMSM$_{RB}(CAS.)$, suggesting that MTL is essential in enhancing downstream task performance. (2) CLMSM$_{RB}$ outperforms all the variants on average score, and 2 out of 5 variations on all 3 metrics. It falters on Cat.3 score. The impact of MSM seems to be high on determining Cat.3 score whereby CLMSM with only MSM or MSM employed *after* CL performs better. (3) CLMSM$_{RB}(RS)$ gives the best Cat. 3 Score among the CLMSM Variations and CLMSM$_{RB}$. This might be attributed to random span masking being helpful in extracting

| | TEST CAT. 1 SCORE | TEST CAT. 2 SCORE | TEST CAT. 3 SCORE | AVERAGE SCORE |
|---|---|---|---|---|
| Rule-based | 57.14 | 20.33 | 2.4 | 26.62 |
| Feature-based | 58.64 | 20.82 | 9.66 | 29.71 |
| ProLocal | 62.7 | 30.5 | 10.4 | 34.53 |
| ProGlobal | 63 | 36.4 | 35.9 | 45.1 |
| EntNet | 51.6 | 18.8 | 7.8 | 26.07 |
| QRN | 52.4 | 15.5 | 10.9 | 26.27 |
| KG-MRC | 62.9 | 40.0 | 38.2 | 47.03 |
| NCET | 73.7 | 47.1 | 41.0 | 53.93 |
| DYNAPRO | 72.4 | 49.3 | **44.5** | 55.4 |
| RECIPES$_{RB}$ | 71.33 | 34.11 | 30.86 | 45.43 |
| RECIPES$_{RL}$ | 74.58 | 47.53 | 38.37 | 53.49 |
| CLMSM$_{RB}(CL)$ | 67.8 | 35.62 | 33.61 | 45.68 |
| CLMSM$_{RB}(MSM)$ | 67.09 | 36.83 | 34.03 | 45.98 |
| CLMSM$_{RB}(CAS.)$ | 67.94 | 36.85 | 35.94 | 46.91 |
| CLMSM$_{RB}(RS)$ | 65.4 | 28.57 | 38.25 | 44.07 |
| CLMSM$_{RB}(EASY)$ | 68.22 | 35.7 | 32.1 | 45.34 |
| CLMSM$_{RB}$ | 69.92 | 39.35 | 33.89 | 47.72 |
| CLMSM$_{RL}$ | **77.26** | **54.86** | 38.34 | **56.82** |

Table 3: Results when fine-tuned on the ProPara Dataset - Baselines and CLMSM Variations vs. CLMSM

location spans.

Additionally, we analyze why CLMSM pre-trained on recipes performs well on open-domain tasks based on attention weights *in Section D.5.2 of Appendix.*

**Comparison with LLMs:** We compare with the following LLM baselines (in Table 4) - (1) Open-source LLMs such as Falcon-7B (pre-trained LLM) and Falcon-7B-Instruct (instruction-fine-tuned Falcon-7B) (Penedo et al., 2023) (2) OpenAI's GPT-3.5 (gpt) and GPT-4 (OpenAI, 2023; Eloundou et al., 2023) models. The LLMs are used in a 1-shot and 3-shot In-Context Learning Setting (Dong et al., 2022).

| | TEST CAT. 1 SCORE | TEST CAT. 2 SCORE | TEST CAT. 3 SCORE | AVERAGE SCORE |
|---|---|---|---|---|
| Falcon-7B (1-shot) | 50.42 | 7.11 | 4.79 | 20.77 |
| Falcon-7B (3-shot) | 50.85 | 7.73 | 5.3 | 21.29 |
| Falcon-7B-Instruct (1-shot) | 50.42 | 5.42 | 0.38 | 18.74 |
| Falcon-7B-Instruct (3-shot) | 48.44 | 3.15 | 1.94 | 17.84 |
| GPT-3.5 (1-shot) | 53.25 | 24.66 | 11.37 | 29.76 |
| GPT-3.5 (3-shot) | 62.43 | 34.66 | 15.81 | 37.63 |
| GPT-4 (1-shot) | 57.2 | 31.08 | 17.03 | 35.10 |
| GPT-4 (3-shot) | 73.87 | **57.7** | 26.78 | 52.78 |
| CLMSM$_{RB}$ | 69.92 | 39.35 | 33.89 | 47.72 |
| CLMSM$_{RL}$ | **77.26** | 54.86 | **38.34** | **56.82** |

Table 4: Results on the ProPara Dataset - LLMs vs. CLMSM

Table 4 shows that - (1) even though Falcon-7B and Falcon-7B-Instruct have almost 6x and 14x the number of parameters compared to CLMSM$_{RB}$ and CLMSM$_{RL}$ respectively, perform considerably worse in comparison (2) both CLMSM$_{RB}$ and CLMSM$_{RL}$ still outperform GPT-3.5 in both settings, and GPT-4 in 1-shot setting on all 4 metrics.

(3) when it comes to GPT-4 in a 3-shot setting, CLMSM$_{RL}$ still beats GPT-4 (3-shot) in Cat. 1, Cat. 3, and Average Scores, and is a close second in Cat 2 Score (4) CLMSM is a highly capable model with a well-tailored pre-training scheme, even though its number of parameters and pre-training data is just a small fraction of that of GPT-3.5 and GPT-4.

#### 4.5.4 Performance Analysis: CLMSM and RECIPES$_{RL}$

We present a case study to elaborate the functioning of CLMSM vis-a-vis the most efficient baseline $RECIPES_{RL}$. Also, we present detailed quantitative results elaborating the difference between the two closely competing models.

**Case Study.** Table 5 shows the ground truth annotations and the predictions of CLMSM$_{RL}$ and $RECIPES_{RL}$ on a procedure from the ProPara Dataset that describes the process of the growth of a tree. We can infer that - (1) $RECIPES_{RL}$ tends to predict a location span in the same step as the entity (while the actual location might be present in some other step), which suggests that $RECIPES_{RL}$ is not able to focus on long-range contexts across steps (2) CLMSM is able to better understand long-range contexts. E.g. in the case of the entities "seedling" and "root system", CLMSM correctly predicts the location corresponding to the ground truth "ground" for most steps (unlike $RECIPES_{RL}$), even though "ground" is mentioned only once in the first step of the procedure. This is because CLMSM learns local as well as global procedural contexts during pre-training. Hence, CLMSM performs better than $RECIPES_{RL}$ for tracking entities whose locations are implicit in nature, i.e., the location at the current step requires retention of knowledge from a step far in the past.

**Quantitative Comparison.** We perform a quantitative analysis of the type of predictions made by CLMSM and the $RECIPES_{RL}$ baseline on the ProPara Test set. We found that (1) CLMSM is able to correctly predict the statuses of "non-existence" and "known-location" of an entity $81.37\%$ and $70.1\%$ of the times respectively, which is better than $RECIPES_{RL}$, which does so $77.59\%$ and $67.68\%$ of the times. (2) When the location of the entity is known, CLMSM predicts the span of the location with $49.04\%$ partial match and $37.62\%$ exact match, while $RECIPES_{RL}$ falls behind again, with $48.39\%$ partial match, and $36.17\%$ exact match. Thus, CLMSM performs better than

| Procedure | Ground Truth | | | | CLMSM$_{RL}$ | | | | $RECIPES_{RL}$ | | | |
|---|---|---|---|---|---|---|---|---|---|---|---|---|
| | seedling | root system | tree | material for new growth | seedling | root system | tree | material for new growth | seedling | root system | tree | material for new growth |
| (Before the process starts) | ground | - | - | - | - | - | - | - | - | - | - | - |
| 1. A seedling is grown in the ground. | ground | - | - | - | ground | - | - | - | ground | - | - | - |
| 2. The seedling grows bigger, and forms its root system. | ground | ground | - | - | ground | ground | - | - | - | seed | - | - |
| 3. The roots start to gain more nourishment as the tree grows. | ground | ground | ground | - | ground | ground | ground | - | - | tree | ground | - |
| 4. The tree starts to attract animals and plants that will grant it more nourishment. | ground | ground | ground | - | ground | ground | ground | - | - | ground | ground | - |
| 5. The tree matures. | ground | ground | ground | - | - | ground | ground | - | - | ground | ground | - |
| 6. After a number of years the tree will grow old and die, becoming material for new growth. | ground | ground | - | ground | - | ground | - | tree | - | ground | - | tree |

Table 5: Analysis of the ground truth and the predictions of CLMSM vs. a well-performing baseline on a sample from the ProPara Dataset. CLMSM is able to predict the ground truth location "ground" in most steps, even though "ground" is mentioned only once in the first step

$RECIPES_{RL}$ in cases where the entity's location is known, as well as, when the entity does not exist.

## 5 Experimental Details: Aligning Actions

As mentioned in Donatelli et al. (2021), action alignment implies each action in one recipe being independently assigned to an action in another similar recipe or left unassigned. Given two similar recipes $R_1$ (source recipe) and $R_2$ (target recipe), we need to decide for each step in $R_1$ the aligning step in the $R_2$, by mapping the word token corresponding to an action in $R_1$ to one or more word tokens corresponding to the same action in $R2$. However, some actions in the source recipe may not be aligned with actions in the target recipe.

An example of the task is explained as follows - Let $R_1$ and $R_2$ be recipes to make waffles. We observe the alignment of the step *"**Ladle** the batter into the waffle iron and **cook** [...]"* in $R_1$ with the step *"**Spoon** the batter into **preheated** waffle iron in batches and **cook** [..]"* in $R_2$ (Words in **bold** are the actions). The relationship between the specific actions of "Ladle" in $R_1$ and "Spoon" in $R_2$ as well as between both instances of "cook" may not be easy to capture when using a coarse-grained sentence alignment, which makes the task challenging.

The details about fine-tuning on the task are described *in Section E of Appendix.*

### 5.1 Aligned Recipe Actions (ARA) Dataset

The ARA dataset (Donatelli et al., 2021) consists of same recipes written by different authors, thus the same recipe might have different number of steps. Thus the recipes in this dataset are lexically different, however semantically similar. The dataset is an extension of crowdsourced annotations for

action alignments between recipes from a subset of (Lin et al., 2020) dataset. The dataset includes 110 recipes for 10 different dishes, with approximately 11 lexically distinct writeups for each dish. These recipes have been annotated with 1592 action pairs through crowdsourcing. 10 fold cross-validation is performed, with recipes of 1 dish acting as the test set each time.

### 5.2 Results

We compare CLMSM$_{BERT}$ with the baseline **RECIPES$_{BERT}$** (BERT-BASE-UNCASED pre-trained on the pre-training corpus of recipes using MLM) and CLMSM Variations. Note that we use models with BERT-BASE-UNCASED architecture similar to Donatelli et al. (2021), which proposed the task. The evaluation (test accuracy) is based on the fraction of exact matches between the ground truth and the predicted action tokens on the test set. The results are shown in Table 6.

**Comparison with Baseline:** CLMSM$_{BERT}$ performs significantly better ($p < 0.05$), showing an improvement of about $4.28\%$ over the baseline **RECIPES$_{BERT}$**. This considerable improvement is obtained intuitively due to CL on hard triplets (capturing similarity between entities of the recipes helps in aligning actions across recipes) and MSM (predicting a masked step given other steps implicitly helps learn similarity of actions across different steps).

**Comparison with CLMSM Variations:** We can infer that - (1) CLMSM$_{BERT}$ outperforms all the variations, as using an MTL framework on the proposed pre-training objectives enhances downstream action alignment performance, and (2) CLMSM$_{BERT}(MSM)$ performs the best among all the variations, as MSM helps capture step-wise

procedural context, helping in action alignment.

|  | TEST ACCURACY |
|---|---|
| RECIPES$_{\text{BERT}}$ | 64.22 |
| CLMSM$_{\text{BERT}}(CL)$ | 55.96 |
| CLMSM$_{\text{BERT}}(MSM)$ | 61.47 |
| CLMSM$_{\text{BERT}}(CAS.)$ | 55.96 |
| CLMSM$_{\text{BERT}}(RS)$ | 54.31 |
| CLMSM$_{\text{BERT}}(EASY)$ | 57.06 |
| CLMSM$_{\text{BERT}}$ | **66.97** |

Table 6: Results when fine-tuned on the ARA Dataset - Baselines vs. CLMSM

## 6 Summary and Conclusion

In this paper, we propose CLMSM, a novel pre-training framework specialized for procedural data, that builds upon the dual power of Contrastive Learning and Masked Step Modeling. The CL framework (with hard triplets) helps understand the identities (entities and actions) which drive a process, and MSM helps to learn an atomic unit of a process. CLMSM achieves better results than the state-of-the-art models in various procedural tasks involving tracking entities in a procedure and aligning actions between procedures. More importantly, even though CLMSM is pre-trained on recipes, it shows improvement on an open-domain procedural task on ProPara Dataset, thus showing its generalizability. We perform an extensive ablation study with various possible variants and show why the exact design of CL and MSM (within the pre-training framework) is befitting for the tasks at hand. We believe that our proposed solution can be extended to utilize pre-training data from multiple domains containing procedural text.

## Limitations

Our work performs pre-training on procedural text in the domain of recipes. Metadata other than recipe names (e.g. cuisine) could be used for sampling similar and dissimilar recipes. This work can be extended to other domains containing procedural text, however, that may bring in new challenges which we would need to solve. For example, for the text from the scientific domain, E-Manuals, etc. that contain a mixture of factual and procedural text, an additional pre-processing stage of extracting procedural text needs to be devised.

## Ethics Statement

The proposed methodology is, in general, applicable to any domain containing procedural text. Specifically, it can potentially be applied to user-generated procedures available on the web and is likely to learn patterns associated with exposure bias. This needs to be taken into consideration before applying this model to user-generated procedures crawled from the web. Further, like many other pre-trained language models, interpretability associated with the output is rather limited, hence users should use the output carefully.

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

# Appendix

The Appendix is organized in the same sectional format as the main paper. The additional material of a section is put in the corresponding section of the Appendix so that it becomes easier for the reader to find the relevant information. Some sections and subsections may not have supplementary material so only their name is mentioned.

## A  Introduction

## B  Pre-training

### B.1  Contrastive Learning using procedure similarity (CL)

### B.2  Mask-Step Modeling (MSM)

### B.3  CLMSM Framework

### B.4  Pre-training in the Recipe Domain

**Source of the pre-training data**

1. **Recipe1M+ dataset** (Marin et al., 2018): This dataset consists of 1 million textual cooking recipes extracted from popularly available cooking websites such as cookingpad.com, recipes.com, and foodnetwork.com amongst others. They include recipes of all the cuisines unlike (Chen and Ngo, 2016) which only contain recipes of Chinese cuisine.

2. **RecipeNLG dataset** (Bień et al., 2020): This dataset was further built on top of Recipe1M+ dataset. Deduplication of the dataset was carried out in order to remove similar samples using the cosine similarity score, which was calculated pairwise upon the TF-IDF representation of the recipe ingredients and instructions. In the end, the dataset added an additional 1.6 million recipes to our corpus.

3. (Majumder et al., 2019): They collect recipes along with their user reviews for a duration of the past 18 years and further filter them using a certain set of rules to create a dataset of about 180,000 recipes and 700,000 reviews which could be used as recipe metadata in future experimentation.

4. (Chandu et al., 2019) : This dataset was scraped from how-to blog websites such as instructables.com and snapguide.com, comprising step-wise instructions of various how-to activities like games, crafts, etc. This a relatively small dataset comprising of around 33K recipes.

| Dataset | Number of Recipes | Number of words | |
|---|---|---|---|
| | | (only steps) | (only ingredients) |
| Recipe1M+ | 1,029,720 | 137,364,594 | 54,523,219 |
| RecipeNLG | 1,643,098 | 147,281,977 | 73,655,858 |
| (Majumder et al., 2019) | 179,217 | 23,774,704 | 3,834,978 |
| (Chandu et al., 2019) | 33,720 | 26,243,714 | - |
| **Total** | **2,885,755** | **334,664,989** | **132,014,055** |

Table 7: Statistics of Recipe Corpus

Table 7 shows the statistics of the pre-training corpus containing over 2.8 million recipes.

**Modification of a recipe by adding ingredients as an extra step:** An example of such a modification of a recipe is shown in Figure 3.

**Pre-training Setup**

| CLMSM ENCODER ARCHITECTURE | NUMBER OF TRAINABLE PARAMETERS | NUMBER OF GPU-HOURS |
|---|---|---|
| BERT-BASE-UNCASED | 397,089,198 | $\approx 14$ |
| RoBERTa-BASE | 488,126,475 | $\approx 16$ |
| RoBERTa-LARGE | 1,217,495,307 | $\approx 40$ |

Table 8: Number of trainable parameters in CLMSM for different encoder architectures and the corresponding pre-training compute in terms of GPU-HOURS

Table 8 shows the number of trainable parameters in CLMSM for 3 different encoder architectures and the corresponding pre-training compute

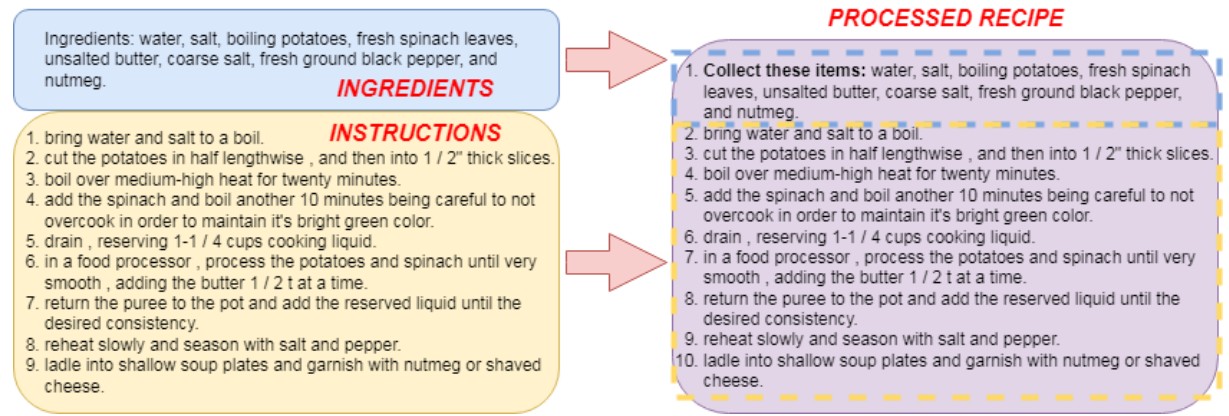

Figure 3: Modifying a recipe by adding ingredients as a step at the top of the original recipe

in terms of GPU-HOURS (number of GPUs used multiplied by the training time in hours).

## C   Experimental Setup

### C.1   CLMSM Variations

## D   Experimental Details: Tracking Entities

### D.1   Problem Definition: Tracking Entity

### D.2   Fine-tuning

**Intermediate Fine Tuning on SQuAD Dataset:** The SQuAD 2.0 dataset (Rajpurkar et al., 2018) is a span-based open-domain reading comprehension dataset that consists of $130,319$ training, $11,873$ dev, and $8,862$ test QA pairs. To improve performance on downstream tasks, the encoder is fine-tuned on the SQuAD 2.0 training set before being fine-tuned on the task-specific dataset. The hyperparameters used in this process are the same as those mentioned in Rajpurkar et al. (2018).

**Hyperparameters for NPN-Cooking Dataset:** The hyperparameters used for fine-tuning are the same as that used in the default implementation of (Faghihi and Kordjamshidi, 2021) - SGD optimizer (Paszke et al., 2017) with $1 \times 10^{-6}$ as the learning rate and a scheduler with a coefficient of $0.5$ to update the parameters every $10$ steps.

**Hyperparameters for ProPara Dataset:** The hyperparameters used for fine-tuning are same as that used in the default implementation of (Faghihi and Kordjamshidi, 2021), - SGD optimizer (Paszke et al., 2017) with $3 \times 10^{-4}$ as the learning rate and a scheduler with a coefficient of $0.5$ to update the parameters every $50$ steps.

### D.3   Common Baselines

**KG-MRC** uses a dynamic knowledge graph of entities and predicts locations by utilizing reading comprehension models and identifying spans of text. **DYNAPRO** is an end-to-end neural model that predicts entity attributes and their state transitions. It uses pre-trained language models to obtain the representation of an entity at each step, identify entity attributes for current and previous steps, and develop an attribute-aware representation of the procedural context. It then uses entity-aware and attribute-aware representations to predict transitions at each step.

### D.4   Entity Tracking on NPN-Cooking Dataset

#### D.4.1   Dataset-specific Baselines

**NPN-Model** converts a sentence to a vector using a GRU-based sentence encoder. The vector representation is then used by an action selector and entity selector to choose the actions and entities in the step. A simulation module applies the chosen actions to the chosen entities by indexing the action and entity state embeddings. Finally, the state predictors make predictions about the new state of the entities if a state change has occurred.

### D.5   Entity Tracking on ProPara Dataset

#### D.5.1   Dataset-specific Baselines

**ProComp** uses rules to map the effects of each sentence on the world state, while the **feature-based** method uses a classifier to predict the status change type and a CRF model to predict the locations. **Pro-Local** makes local predictions about state changes and locations based on each sentence and then globally propagates these changes using a persistence rule. **ProGlobal** uses bilinear attention over the

entire procedure and distance values to make predictions about state changes and locations, taking into account the previous steps and current representation of the entity. **EntNet** consists of a fixed number of memory cells, each with its own gated recurrent network processor that can update the cell's value based on input. Memory cells are intended to represent entities and have a gating mechanism that only modifies cells related to the entities. **QRN** is a single recurrent unit that simplifies the recurrent update process while maintaining the ability to model sequential data. It treats steps in a procedure as a sequence of state-changing triggers and transforms the original query into a more informed query as it processes each trigger. **NCET** uses entity mentions and verb information to predict entity states and track entity-location pairs using an LSTM. It updates entity representations based on each sentence and connects sentences with the LSTM. To ensure the consistency of its predictions, **NCET** uses a neural CRF layer over the changing entity representations.

### D.5.2 Reason behind domain-generalizability

We observe the attention weights between the question and the steps of the open-domain as well as in-domain procedures for the transformer obtained after fine-tuning CLMSM on the open-domain ProPara Dataset.

Figure 4 shows the attention weights. Figure 4a shows the attention weights in layer 19 of the transformer corresponding to tracking the location of the entity "water" in the first two steps of photosynthesis, taken from the open-domain ProPara Dataset. Figure 4b shows the attention weights for the same layer and attention head corresponding to tracking the location of the entity "water" in the first two steps of the Orange Juice recipe, which is an in-domain procedure.

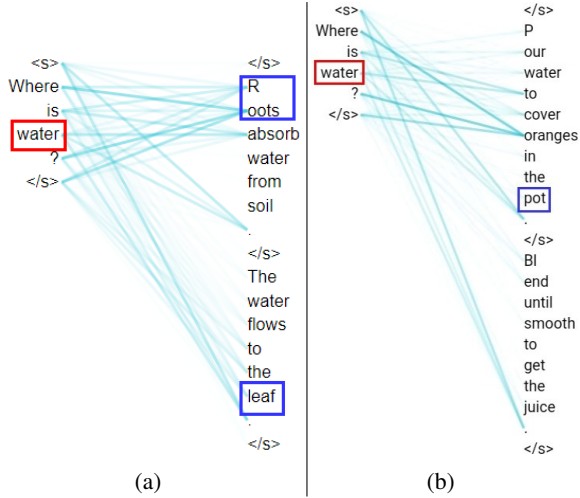

Figure 4: Attention weights between question and procedure steps for Layer 19, Attention Head 8. Self-attention is represented as lines connecting tokens, and the line weights reflect attention scores

From Figure 4, we can infer the following similarities between the open-domain and in-domain scenarios - When considering an inner transformer layer in between (here Layer 19), the question shows high attention towards a location or an entity closer to the beginning of the procedure. For instance, in the open-domain case, the question attends highly to the location "Roots", while in the in-domain case, it attends highly to the entity "oranges".

Hence, this example shows that there is a similarity in the attention maps between the open-domain and in-domain cases, which leads to improved performance in both cases. This similarity might stem from the similarity in the structures of procedures across domains, which could be a potential direction of future work.

## E   Experimental Details: Aligning Actions

**Fine-tuning:** This is treated as an alignment problem where each action in the source recipe is independently aligned to an action in the target recipe or left unaligned. The task uses a two-block architecture with an encoder that generates an encoding vector for each action in the recipe and a scorer that predicts the alignment target for each action using one-versus-all classification. The encoder is an extended model that incorporates structural information about the recipe graph. The scorer uses a multi-layer perceptron to compute scores for the alignment of each source action to every target action, including a "None" target represent-

ing no alignment. The model is trained using a cross-entropy loss function with updates only on incorrect predictions to avoid overfitting.

**Hyperparameters for ARA Dataset:** The hyperparameters used are the ones mentioned in the default implementation of Donatelli et al. (2021).

### E.1 Aligned Recipe Actions (ARA) Dataset

### E.2 Results

## F Summary and Conclusion