# OpenReview forum: "$\textbf{\emph{CLMSM}}$: A Multi-Task Learning Framework for Pre-training on Procedural Text"
_EMNLP/2023/Conference — EMNLP 2023 Findings_

### Official Review · Reviewer_uD45 · 2023-07-27

**Soundness:** 4

**Excitement:**

3: Ambivalent: It has merits (e.g., it reports state-of-the-art results, the idea is nice), but there are key weaknesses (e.g., it describes incremental work), and it can significantly benefit from another round of revision. However, I won't object to accepting it if my co-reviewers champion it.

**Paper Topic And Main Contributions:**

The paper proposes a pretraining framework for the tasks of (1) entity tracking and (2) action aligment in a procedural text. The paper focuses on the recipe domain (NPN-Cooking Dataset; Bosselut et al., 2017) with additional experiments on an open-domain dataset (ProPara Dataset; Mishra et al., 2018). Each example in the datasets is a procedure comprising of multiple steps. The model is required to (1) track the location and status of entities mentioned in the text (e.g., ingredients) or (2) align semantically equivalent steps in two similar procedures.

The pretraining framework proposed by the authors builds on a Triplet Network (Cohan et al., 2020; Ostendorff et al., 2022) with a BERT (Devlin et al., 2018) or RoBERTa (Liu et al., 2019) encoder. The loss for the network is a sum of two objectives: contrastive learning with positive and negative examples (in which the model is required to minimize the distance between representations of positive examples and vice versa), and mask-step modelling (in which the model is required to infill masked tokens of randomly masked step(s) in the procedure).

The authors conduct extensive experiments for both datasets and tasks, including multiple baselines and ablations of the proposed pretraining scheme. The results show the proposed approach is able to (significantly) outperform state-of-the-art baselines and generalize to open-domain tasks.

**Questions For The Authors:**

A) Can you please provide more details on how you arrived at the values of statistical significance stated in the paper?

**Reasons To Accept:**

- The paper proposes a pretraining scheme for two downstream tasks on procedural text which is able to outperform state-of-the-art baselines.
- The authors conduct an extensive ablation study for each experiment, providing insights regarding contribution of individual parts of the proposed approach.
- Besides measuring the test accuracy, the authors discuss the performance of individual baselines, providing more intuition regarding the outcomes.
- The authors provide an anonymized version of the code used for training the models.

**Reasons To Reject:**

- The proposed approach requires four training stages: general-domain mask language modeling (MLM) pretraining, in-domain MLM pretraining, finetuning on the SQuAD 2.0 dataset (Rajpurkar et al., 2018), and finetuning on the downstream task. This leaves many levels of freedom regarding pretraining data and parameters, which in turn may result in brittleness for some domains or for a different choice of hyperparameters.
- No details are provided regarding which statistical test was used to compute the p-values.

**Reproducibility:**

4: Could mostly reproduce the results, but there may be some variation because of sample variance or minor variations in their interpretation of the protocol or method.

**Reviewer Confidence:**

3: Pretty sure, but there's a chance I missed something. Although I have a good feel for this area in general, I did not carefully check the paper's details, e.g., the math, experimental design, or novelty.

**Typos Grammar Style And Presentation Improvements:**

Although I understand the appeal of Appendix containing all the corresponding sections for easier orientation, I'd suggest using a more standard version which would contain only the necessary sections.

---

> ### Author Rebuttal · Authors · 2023-08-29
>
> **1. The proposed approach requires four training stages: general-domain mask language modeling (MLM) pretraining, in-domain MLM pretraining, finetuning on the SQuAD 2.0 dataset (Rajpurkar et al., 2018), and finetuning on the downstream task. This leaves many levels of freedom regarding pretraining data and parameters, which in turn may result in brittleness for some domains or for a different choice of hyperparameters.**
>
> The stage of general domain MLM is prior to our pre-training - we directly take off-the-shelf models such as RoBERTa-BASE.
> Following this, in-domain pre-training works on a massive amount of diverse pretraining data. Hence, it is less task-specific and less sensitive to hyperparameters (For reference, check Table 3 [Liu et al. 2019], the original paper where RoBERTa was introduced). The set of hyperparameters used is exactly the same as that of pre-training RoBERTa, as mentioned in its original paper. We would also like to mention that there is consistency in result (i.e.robustness) amidst plenty of variation in the architecture, loss function, token masking scheme, choice of triplets while pre-training (as shown in our ablation studies).
> For both the fine-tuning tasks, we use the same set of hyperparameters as their corresponding original papers.
> This shows the results obtained did not have to undergo any hyperparameter tuning and thus inherit the robustness (or brittleness) of the original modules which all other baselines also use.
>
> **2. Can you please provide more details on how you arrived at the values of statistical significance stated in the paper?**
>
> To see if our results are statistically significant, we use the paired T-test [Student. (1908). “The probable error of a mean”. Biometrika, 6(1), 1-25.] to compare the results of our proposed solution with baselines, and observe that the p-score is less than 0.05, which intuitively means that our solution is performing significantly better than the corresponding baselines.

---

### Official Review · Reviewer_9spE · 2023-08-01

**Soundness:** 4

**Excitement:**

2: Mediocre: This paper makes marginal contributions (vs non-contemporaneous work), so I would rather not see it in the conference.

**Paper Topic And Main Contributions:**

This paper tackles the problem of modeling processes described in natural language, such as recipes or e-manuals etc. Specifically, it uses a pretraining objective that consists of two components: contrastive learning and masked step modeling. In contrastive learning, a hard input selection isused to teach the model to distinguish between identical vs. similar processes. Meanwhile, masked step modeling (much like masked language modeling), is used to train the model in capturing step-wise context of a process. The authors conduct a series of experiments on a couple of task specific problems, demonstrating that their models outperform several baselines.

**Reasons To Accept:**

Overall, the paper has a rigorous set of experiments and does a thorough job of testing the proposed model on several benchmarks and in several different settings.

**Reasons To Reject:**

Unfortunately, a few issues prevent me from recommending this paper.

My biggest gripe is with the novelty and relevance of the paper. The method appears to be a combination of previously proposed approaches, including triplet networks and masked step modeling for procedural video understanding. This isn't, in and of itself, a disqualifying factor, but combined with a generally weak motivation it isn't clear why the community would benefit from adopting this method (or why modeling procedural texts is useful to begin with). Finally, combine this with the fact that the authors do not compare their approach to an off-the-shelf LLM, whether in zero-shot or few-shot setting, means that there is no way of knowing whether these findings (even if they appear to improve on some baselines), continues to be still relevant in today's day and age.

More generally, while the paper isn't poorly written, it is very dense. The liberal use of opaque acronyms, the general absence of helpful examples, the lack of clear definitions for the input units and outputs of the model, and interpretive commentary that fails to motivate why this is an important problem or why the proposed solution is a good one, means that it is a generally difficult paper to parse. Overall, I am left with an impression of an interesting model with some nice results but not much more.

**Reproducibility:**

2: Would be hard pressed to reproduce the results. The contribution depends on data that are simply not available outside the author's institution or consortium; not enough details are provided.

**Reviewer Confidence:**

4: Quite sure. I tried to check the important points carefully. It's unlikely, though conceivable, that I missed something that should affect my ratings.

---

> ### Author Rebuttal · Authors · 2023-08-29
>
> **1. Why modeling procedural texts is useful to begin with?**
>
> In this work, we explore fundamental tasks within procedural reasoning - tracking (1) entities and (2) actions acted upon those entities, across the steps of a procedure. Automated tracking of entities in a complex procedure helps an end user understand the life cycle of different entities, along with the extent to which an entity participates/is acted upon in the process. We get a glimpse of this when tracking entity locations in the Propara Dataset, and this notion can be extended to even more complex biological and chemical processes. Secondly, aligning actions across procedures (written by different people) helps an end-user understand similarities and the exact subtle differences across the procedures that ultimately have the same goal. For instance, two Biryani recipes could have many common but subtly different cooking actions ("saute vegetables" vs. "fry vegetables"), giving two very similar dishes, having subtle differences in tastes.
>
> From the perspective of NLP, Procedural Reasoning is challenging, as entity states and actions across steps of a procedure might not be explicitly mentioned in a step/could require information present in a past/future step (see "Case Study" in Section 4.5.4). This requires NLP models to have world knowledge as well as commonsense reasoning capability.
>
> Also, reasoning on Procedural text is essential for a myriad of other applications such as - (1) making a tutorial for understanding complex life processes and chemical reactions  (2) Helping robots perform a task step-by-step without any human intervention. (3) Question Answering on Instruction Manuals having instructions for operating different devices to help in troubleshooting and customer support.
>
>  **2. How relevant and novel is the paper?**
>
> **Relevance of the paper** - As answered in the previous question, modeling procedural texts is extremely useful in today’s AI research landscape. Our paper discusses a pre-training pipeline customized to procedural text, making our work relevant to the conference, as well as to the NLP research community as a whole.
>
> **Novelty of the paper** - Our work is the first attempt at proposing a domain-specific, pre-training framework for procedural reasoning. We propose a never-before-tried combination of triplet loss (used in previous works) and masked-step-modeling (MSM is inspired by the notion of masked span modeling, as in SpanBERT, extended to accommodate the step-wise nature of procedures) in our proposed pre-training pipeline. Additionally, we fine-tune and evaluate such a pre-trained model on different datasets on procedural reasoning (for which such pre-training on a large corpus of procedural data has not been explored previously), and observe that our proposed approach beats many competitive baselines. Hence, from the viewpoint of (1) architecture, (2) loss functions, and (3) use of such a pipeline for procedural reasoning, we would argue that our proposed CLMSM is indeed a novel addition to the procedural reasoning paradigm in the research community.
>
> **3. Is the choice of baselines relevant to today’s day and age?**
>
> Yes, the baselines we choose to compare CLMSM with, are indeed recent and competitive. For instance, 9 out of 11 baselines in the case of ProPara are no older than 5 years, and all the baselines have been shown to be relevant to procedural reasoning in several prior art. Also, the pool of baselines we use is diverse, with the following broad categories - (1) architecture specific to the procedural task at hand (E.g. ProLocal, ProGlobal) (2) model infused with external knowledge (e.g. KG-MRC) (3) baselines using pre-trained models (e.g. RECIPES_RB, RECIPES_RL). Hence, we feel that our choice of baselines is also comprehensive.
>
> **4. Did you compare CLMSM to off-the-shelf LLM in zero-shot or few-shot setting**?
>
> CLMSM is compared to several open-domain and domain-specific LLMs such as RoBERTa-BASE, RoBERTa-LARGE, BERT-BASE, RECIPES_RB, RECIPES_RL, RECIPES_BERT (Note: In the Wikipedia Article on "Large language model", under the "List" Section, BERT has been mentioned. Since RoBERTa has a similar architecture, it is also therefore an LLM. Subsequently, continually pre-training such a model would also result in an LLM). Since our work is the first work on domain-specific pre-training in the procedural reasoning paradigm, we keep the fine-tuning setup the same as the one followed by the corresponding dataset papers, to ensure equitable and fair comparison between CLMSM and other baselines.
>
> However, for completeness, we are adding some results of larger, instruction-tuned LLMs such as Falcon-7B (pre-trained LLM) and Falcon-7B-Instruct (instruction-fine-tuned Falcon-7B) [Penedo et al 2023] on the ProPara Dataset. Also, we use the few-shot (in-context learning) setting, as LLMs have shown remarkable results in such settings in a variety of NLP tasks.
>
> |  | Cat 1 Accuracy | Cat 2 Accuracy | Cat 3 Accuracy |
> | - | - | - | - |
> | Falcon-7B (1-shot) | 50.42 | 7.11 | 4.79 |
> | Falcon-7B (3-shot) | 50.85 | 7.73  |  5.3  |
> | Falcon-7B-Instruct (1-shot) | 50.42 | 5.42 | 0.38 |
> | Falcon-7B-Instruct (3-shot) | 48.44 | 3.15  | 1.94   |
> | CLMSM_RB | 69.92 | 39.35 | 33.89 |
> | CLMSM_RL | **77.26** | **54.86** | **38.34** |
>
> We observe that such an LLM, which has almost 6x and 14x the number of parameters compared to CLMSM_RB and CLMSM_RL, performs considerably worse in comparison. Thus, in-context learning using very large LLMs, which work in several NLP tasks, fails to match up with the performance of smaller fine-tuned models, such as our solution. This also goes on to show that the problem of procedural reasoning is indeed non-trivial. We consider the suggestion and would explore the performance of large LLMs on procedural reasoning, and potential ways to improve performance.
>
> **5. Paper is very dense. The liberal use of opaque acronyms, the general absence of helpful examples, the lack of clear definitions for the input units and outputs of the model, and interpretive commentary that fails to motivate why this is an important problem or why the proposed solution is a good one, means that it is a generally difficult paper to parse.**
>
> We would like to address each concern of the reviewer (regarding the writing of the paper) as follows -
>
> - **Use of opaque acronyms** - The meaning/full-form of all acronyms (e.g. CL, MSM, MTL, etc.) used in the paper are written when they are defined for the first time, and most of the acronym expansions can be seen in the Introduction. For later usages of the acronym, we do not mention the full form any more to conserve the available paper space. In the final version, we will provide a table with expansion of all the acronyms for ready reference.
>
> - **General absence of helpful examples** - Table 1, Table 4 show examples of open-domain processes, and Figure 3 in the Appendix shows an example recipe. However, as per the reviewer’s suggestion we would add some example recipes in the main paper for the benefit of the reader.
>
> - **lack of clear definitions for the input units and outputs of the model** -
>
> | Training Phase | Task/Objective | Input Units | Output Units | Reference in  paper |
> | - | - | - | - | - |
> | Pre-training | Contrastive Learning | sampled anchor, positive, and negative (step-masked) procedures | procedure representations, which are later utilized in calculating the triplet margin loss | Section 2.1, Fig. 1 |
> | Pre-training | Mask-Step Modeling | Procedure with masked step tokens | predicted mask tokens, which are learned via the MSM Loss | Section 2.2, Fig. 1 |
> | Fine-tuning | Entity Tracking | entity, step of the process | State of the entity | last paragraph of Section 4.1 |
> | Fine-tuning | Action Alignment | similar recipes | Aligned Actions across steps of the recipes | first paragraph of Section 5, and the same is explained through the use of an example in the following paragraph |
>
>
> a. **During pre-training -** Sampled anchor, positive, and negative (step-masked) procedures are inputs, and the procedure representations are the outputs, which are later utilized in calculating the triplet margin loss in Equation 1 (this is clearly mentioned in Section 2.1 and in Fig. 1). Also, procedure with masked step tokens is input, and the outputs are the predicted mask tokens, which are learned via the MSM Loss (see Section 2.2 and Fig. 1).
>
> b. **During Fine-tuning -** for tracking entities, the inputs (entity and step of the process) and outputs (state of the entity) are formally defined in the last paragraph of Section 4.1; for aligning actions, the inputs (similar recipes) and outputs (Aligned Actions across steps of the recipes) are formally defined in the first paragraph of Section 5, and the same is explained through the use of an example in the following paragraph.
>
> - **interpretive commentary that fails to motivate why this is an important problem or why the proposed solution is a good one** - The answer to the first question shows why this is an important problem. The answer to the second question shows why the proposed CLMSM is novel. Also, Sections 4.4.3, 4.5.3, and 5.2 show that CLMSM outperforms relevant and competitive baselines. Hence, the proposed solution is satisfactory in our opinion.

---

### Official Review · Reviewer_z7wU · 2023-08-05

**Soundness:** 4

**Excitement:**

3: Ambivalent: It has merits (e.g., it reports state-of-the-art results, the idea is nice), but there are key weaknesses (e.g., it describes incremental work), and it can significantly benefit from another round of revision. However, I won't object to accepting it if my co-reviewers champion it.

**Paper Topic And Main Contributions:**

This paper proposes a multi-task learning framework for continual training on procedural data, which involves both Contrastive Learning using procedure similarity and Mask-Step Modeling. The authors evaluate this framework on recipe data and achieve good performance and generalization performance.

**Reasons To Accept:**

- This paper presents a multi-task framework for pretraining/continual training on procedural data and outperforms the baseline, which is the state-of-the-art method when this work is done.

- The authors conducted extensive and comprehensive experiments, oblation study the effectiveness of the proposed framework.

**Reasons To Reject:**

- The writing of this paper requires improvement. The authors could add one or more examples to demonstrate the procedural data the model is trained on. It is not easy for people unfamiliar with the recipe data to get a good intuition of this method. Also, many sections seem a little bulky, like lines 435-468.

**Reproducibility:**

4: Could mostly reproduce the results, but there may be some variation because of sample variance or minor variations in their interpretation of the protocol or method.

**Reviewer Confidence:**

3: Pretty sure, but there's a chance I missed something. Although I have a good feel for this area in general, I did not carefully check the paper's details, e.g., the math, experimental design, or novelty.

---

> ### Author Rebuttal · Authors · 2023-08-29
>
> **1. The authors could add one or more examples to demonstrate the procedural data the model is trained on. It is not easy for people unfamiliar with the recipe data to get a good intuition of this method.**
>
> Table 1, Table 4 show examples of open-domain processes, and Figure 3 in Appendix shows an example recipe. However, we acknowledge the reviewer’s comment and would add some example recipes in the main paper for the benefit of the reader.
>
> **2. Also, many sections seem a little bulky, like lines 435-468**
>
> From lines 435-468, we have compared the results between our method and baselines, with a reasoning behind the relative difference in model performance. In order to fit such content within the confines of available space, perhaps the content has become dense. However, as per the reviewer’s suggestion, we would carefully review the entire paper and ensure that the content becomes more lucid.

---

### Meta-Review · Area_Chair_NtWc · 2023-09-19

**Recommendation:** 3

**Metareview:**

The paper proposes a multi-task learning framework for continual training on procedural data, combining the methods of contrastive learning and mask-step modeling.

The paper contains a thorough set of experiments and the final system achieves state-of-the-art results.
There is also sufficient analysis and ablation experiments.

As a downside, the framework seems to be a combination of two existing methods, which limits its novel contribution.
Relevance compared to zero-shot LLMs is left somewhat unclear.
Several reviewers point out that the paper is difficult to read and understand, containing dense text, many acronyms, and very few examples.

---

### Decision · Program_Chairs · 2023-10-07

**Decision:**

Accept-Findings

**Comment:**

The paper proposes a multi-task learning framework for continual training on procedural data, combining the methods of contrastive learning and mask-step modeling.

The paper contains a thorough set of experiments and the final system achieves state-of-the-art results.
There is also sufficient analysis and ablation experiments.

As a downside, the framework seems to be a combination of two existing methods, which limits its novel contribution.
Relevance compared to zero-shot LLMs is left somewhat unclear.
Several reviewers point out that the paper is difficult to read and understand, containing dense text, many acronyms, and very few examples.